# Susceptibility of Fat Tissue to SARS-CoV-2 Infection in Female hACE2 Mouse Model

**DOI:** 10.3390/ijms24021314

**Published:** 2023-01-09

**Authors:** Hariprasad Thangavel, Dhanya Dhanyalayam, Kezia Lizardo, Neelam Oswal, Enriko Dolgov, David S. Perlin, Jyothi F. Nagajyothi

**Affiliations:** Center for Discovery and Innovation, Hackensack Meridian Health, Nutley, NJ 07110, USA

**Keywords:** SARS-CoV-2, COVID-19, adipose tissue, adipocytes, hACE2 mice, immune signaling, inflammatory cytokines, fat loss, apoptosis, cell death

## Abstract

The coronavirus disease (COVID-19) is a highly contagious viral illness caused by severe acute respiratory syndrome coronavirus-2 (SARS-CoV-2). COVID-19 has had a catastrophic effect globally causing millions of deaths worldwide and causing long-lasting health complications in COVID-19 survivors. Recent studies including ours have highlighted that adipose tissue can act as a reservoir where SARS-CoV-2 can persist and cause long-term health problems. Here, we evaluated the effect of SARS-CoV-2 infection on adipose tissue physiology and the pathogenesis of fat loss in a murine COVID-19 model using humanized angiotensin-converting enzyme 2 (hACE2) mice. Since epidemiological studies reported a higher mortality rate of COVID-19 in males than in females, we examined hACE2 mice of both sexes and performed a comparative analysis. Our study revealed for the first time that: (a) viral loads in adipose tissue and the lungs differ between males and females in hACE2 mice; (b) an inverse relationship exists between the viral loads in the lungs and adipose tissue, and it differs between males and females; and (c) CoV-2 infection alters immune signaling and cell death signaling differently in SARS-CoV-2 infected male and female mice. Overall, our data suggest that adipose tissue and loss of fat cells could play important roles in determining susceptibility to CoV-2 infection in a sex-dependent manner.

## 1. Introduction

COVID-19 is a viral respiratory illness, caused by severe acute respiratory syndrome coronavirus-2 (SARS-CoV-2) [1]. It causes debilitating disease manifestations in many infected people and increases mortality in people with comorbidities, including metabolic disorders and heart diseases [2,3,4,5,6,7,8]. The causes of death in COVID-19 patients include cardiomyopathy, stroke, cardiac arrest, sepsis, and organ failure [9,10,11,12,13,14]. At least 50% of COVID-19 survivors are known to face lingering health issues, which include a racing heartbeat, shortness of breath, achy joints, and damage to the heart, lungs, kidney, and brain [15,16]. A recent meta-analysis review that included 47,910 patients (age 17–87 years) estimated the prevalence of 55 long-term post-COVID-19 effects, where 58% of patients suffer from fatigue, 12% of which is due to significant weight loss [17]. Other clinical reports suggest that the post-COVID-19 stage is associated with acute (30%) and chronic weight loss (56%) and malnutrition [18]. Loss of body weight is linked with body fat mass and the pathophysiology of fat cells. Adipocytes, also known as fat cells, regulate inflammatory signaling and immune response [19,20,21,22,23,24]. It is well known that body fat levels and distribution patterns differ between the sexes and races [25,26,27], which may influence the susceptibility to SARS-CoV-2 infection, COVID-19-associated symptoms, and side effects. Importantly, recent studies have shown that SARS-CoV-2 infects adipose tissue [28,29].

In the present pilot study, we investigated the effect of SARS-CoV-2 infection on adipose tissue physiology and the pathogenesis of fat loss in a murine model of COVID-19 using humanized angiotensin-converting enzyme 2 (hACE2) mice. We used both male and female hACE2 mice intra-nasally infected with SARS-CoV-2. We demonstrated that CoV-2 infects adipose tissue and persists around the lipid droplets in white adipose tissue (WAT) of CoV-2-infected mice 10 days post infection (DPI). Our studies revealed that in male and female mice, CoV-2 infection differently affects adipose tissue and regulates immune signaling. Thus, the alterations in adipose tissue metabolic and immunologic functions may affect the whole-body immune and metabolic homeostasis differently in males and females during acute COVID-19 illness and the post-COVID-19 phase. These data may help explain the higher COVID-19 susceptibility in males compared to females.

## 2. Results

Earlier, we demonstrated that SARS-CoV-2 infection alters pulmonary pathology in hACE2 mice differently in males and females [30]. In particular, we showed a significantly increased viral load and infiltration of immune cells in the lungs of infected male mice compared to female mice at 10 DPI [30]. However, both male and female mice showed decreased body weight compared to control mice. Our earlier studies suggest that decreased body weight is likely caused by a loss of body fat mass [30]. Therefore, we investigated the role of adipose tissue in CoV-2 infection using white adipose tissue (WAT) of infected and uninfected control mice (10 DPI) as detailed in the Materials and Methods section. To investigate the pathological effects of SARS-CoV-2 infection in the WAT of hACE2 mice, we performed histological and biochemical analyses of WAT samples at 10 DPI. Age and sex-matched uninfected mice served as controls. We used *n* = 8 mice/group (4 uninfected and 4 CoV-2 infected) for both sexes. We observed no mortality during CoV-2 infection in mice up to and including 10 DPI. However, the histological analysis of WAT samples has revealed a significant difference in their pathology between the sexes. Therefore, we analyzed all data separately for males and females as presented below.

### 2.1. SARS-CoV-2 Infection Alters Adipose Tissue Morphology Differently in Male and Female hACE2 Mice

Histological analysis of WAT was performed using H&E (Figure 1a) and Masson-trichrome (Figure 1b) stained sections as described in Materials and Methods. Microscopic analysis of the histological sections of WAT demonstrated significantly increased levels of infiltrating immune cells, loss of lipid droplets, and evidence of increased fibrosis in CoV-2 infected hACE2 mice compared to uninfected mice (Figure 1). Between uninfected male and female mice, the size of adipocytes was relatively larger in females (Figure 1a). However, female mice lost a significant amount of body fat compared to males during CoV-2 infection (Figure 1a) [31,32]. We observed increased fibrosis in adipose tissue in infected mice compared to uninfected mice (Appendix A). These data suggest that adipose tissue undergoes significant morphological changes, including increased immune cell infiltration and loss of lipid droplets, which can alter the local and systemic immune and metabolic homeostasis during CoV-2 infection.

### 2.2. Sex Differences in the Tissue CoV-2 Tropism in the Lungs and Visceral Fat Pads

ACE2 protein is a well-recognized receptor for CoV-2 entry into the host cell [33,34]. Earlier we showed by Western blotting that CoV-2 infection increases the expression levels of ACE2 protein in the lungs of hACE2 mice [30]. The levels of ACE2 were significantly higher in the lungs of both male (*p* ≤ 0.0001) and female (*p* ≤ 0.01) mice infected with SARS-CoV-2 compared to sex-matched uninfected (control) mice [30]. Here, we analyzed whether CoV-2 infection also alters the levels of ACE2 in WAT. In WAT, the levels of ACE2 were significantly higher in male (*p* < 0.05) CoV-2 infected mice compared to sex-matched control mice (Figure 2a). We analyzed the viral loads in the lung and WAT by qPCR analysis. Lung viral loads were significantly greater in male CoV-2-infected mice compared to female CoV-2-infected mice (Figure 2b), which may be due to increased ACE2 levels in male mice [30]. However, qPCR analysis demonstrated significantly higher levels of viral load in the WAT of female CoV-2 infected mice (64-fold, *p* ≤ 0.005) compared to male CoV-2 infected mice, although the levels of ACE2 were not significantly increased (Figure 2c). We also performed immunohistochemistry (IHC) analysis of SARS-CoV-2 using a monoclonal antibody against the SARS-CoV-2 nucleocapsid protein, which demonstrated the presence of SARS-CoV-2 nucleocapsid protein in adipose tissue around the lipid droplets in infected mice (Figure 2d). These data demonstrate that: (i) CoV-2 infection alters ACE2 levels and viral loads differently in male and female mice; (ii) SARS-CoV-2 infects and persists in adipose tissue; (iii) adipose tissue in females may act as a sink/reservoir for CoV-2; and (iv) an inverse relationship exists between the viral loads in the lungs and adipose tissue.

### 2.3. SARS-CoV-2 Infection Alters Immune Signaling in the Adipose Tissue Differently in Male and Female hACE2 Mice

Immunoblot analysis of WAT lysates demonstrated significant differences in the protein levels of immune cell markers indicating altered levels of CD4^+^ cells CD8^+^ cells and F4/80^+^ cells; and inflammatory cytokines such as TNFα, IL-6, and IL-10 between the sexes during CoV-2 infection (Figure 3). Uninfected female mice showed significantly lower levels of resident CD4^+^ cells (*p* < 0.05) and CD8^+^ cells (*p* < 0.05) compared to uninfected male mice (Figure 3a). CoV-2 infection significantly increased the infiltration of CD4^+^ and CD8^+^ cells in WAT in both males (*p* < 0.05 and *p* < 0.005, respectively) and females (*p* < 0.0001 and *p* < 0.0001, respectively) compared to their respective sex-matched uninfected mice. The levels of F4/80^+^ cells in WAT were significantly elevated (*p* < 0.01) in the WAT of CoV-2 infected mice compared to uninfected mice, irrespective of their sex (Figure 3a). Overall, the levels of CD4^+^ cells and CD8^+^ cells) were significantly increased (*p* < 0.0001) in the WAT of female CoV-2 mice compared to male CoV-2 mice.

There was no significant difference between the levels of proinflammatory TNFα in the WAT of male and female control mice (Figure 3b). However, CoV-2 infection significantly increased the levels of TNFα in female mice compared to their sex-matched controls (*p* < 0.01) and infected male counterparts (*p* < 0.05). Similarly, the levels of IL-6 were significantly elevated in CoV-2-infected female mice compared to their male counterparts (*p* < 0.001) and sex-matched controls (*p* < 0.0001). No significant change in the levels of IL-10 was observed in either male or female CoV-2-infected mice compared to the corresponding sex-matched control groups. These data demonstrated that CoV-2 infection induces stronger proinflammatory signaling in the WAT of female mice compared to male mice.

### 2.4. SARS-CoV-2 Infection Alters Immune Signaling in the Adipose Tissue Differently in Male and Female hACE2 Mice

Immunoblot analysis of WAT lysates demonstrated significant differences in the protein levels of lipases (ATGL and p-HSL) between the sexes in hACE2 mice and during CoV-2 infection (Figure 4). Uninfected female mice showed significantly lower levels of ATGL (*p* < 0.01) and p-HSL (*p* < 0.01) expression compared to uninfected male mice (Figure 4). Furthermore, CoV-2 infection significantly increased the levels of ATGL (*p* < 0.05) and p-HSL (*p* < 0.01) in females compared to female uninfected mice. However, the levels of ATGL significantly decreased (*p* < 0.05), and the levels of p-HSL were not altered in infected male mice compared to uninfected male mice. These data indicate that CoV-2 infection differently activates lipases in WAT between male and female hACE2 mice.

### 2.5. SARS-CoV-2 Infection Causes a Loss of Fat Cells in hACE2 Mice

Histological analysis demonstrated a significant loss of lipid droplets and adipocytes in CoV-2 infected mice compared to their control groups (Figure 1). We analyzed whether the cause for the loss of adipocytes was due to apoptosis or necrosis by quantitating the protein levels of cleaved caspase 3 and Bnip3, respectively, in the WAT (Figure 5). In the WAT of uninfected female mice, the levels of cleaved caspase 3 and Bnip3 were slightly elevated compared to uninfected male mice; however, this difference was not statistically significant. In contrast, CoV-2 infection significantly increased the levels of cleaved caspase 3 (*p* < 0.0001) (Figure 5a) and Bnip3 (*p* < 0.001) (Figure 5b) in females compared to uninfected controls. Interestingly, the levels of cleaved caspase 3 in the WAT of CoV-2 females were significantly higher (*p* < 0.0001) compared to their male counterparts, suggesting that cell death in the WAT of female mice may be predominantly driven by apoptosis (Figure 5a). In addition, the levels of necrotic cell death markers in the WAT of CoV-2 infected mice were also significantly elevated (*p* < 0.0001) in females compared to males (Figure 5b). These data indicate that during CoV-2 infection adipose tissue is lost via both apoptotic and necrotic cell death in females more so than in males.

## 3. Discussion

The epidemiological findings reported globally indicate higher morbidity and mortality in males than in females with SARS-CoV-2 infection [35,36,37]. Although women also get infected with CoV-2, many clinical studies indicate that males are more susceptible to developing severe COVID-19 compared to females, and many researchers have attributed this difference to sex-specific hormones [38,39,40,41]. A few reports have also suggested a difference in immune responses between the sexes [42,43,44]. However, how exactly the immune response may change between males and females during CoV-2 infection is not well understood. In our previous studies, we have demonstrated that pathogens such as parasites, such as *Trypanosoma cruzi*, and bacteria, such as *Mycobacterium tuberculosis*, can infect and persist in adipose tissue [30,45,46]. Recently, we and others have shown that CoV-2 can also infect adipose tissue [28,30,47,48]. In particular, biopsies have demonstrated the presence of CoV-2 in subcutaneous thoracic fat [28] and abdominal fat [29] in COVID-19 patients. These studies have shown that adipose tissue can be a significant reservoir for SARS-CoV-2 and an important source of inflammatory mediator IFN-γ [29]. The present study investigated sex differences in: (i) SARS-CoV-2 viral loads in adipose tissue; and (ii) immune signaling due to the presence of CoV-2 in adipose tissue. Moreover, this study assessed whether the relationship between lung and adipose tissue viral loads differs between male and female infected hACE2 mice. Our study revealed for the first time that: (a) viral loads in adipose tissue and the lungs differ between males and females; (b) an inverse relationship exists between the viral loads in the lungs and adipose tissue, and it differs between the males and females in hACE2 mice; and (c) CoV-2 infection alters immune signaling, lipolysis, and cell death signaling differently in the adipose tissue of SARS-CoV-2 infected male and female mice.

Earlier we showed that the viral loads in the lungs of female CoV-2 infected mice were significantly lower compared to male CoV-2 infected mice, which is reminiscent of the observations made in COVID-19 patients [30,49]. The increased lung pathology observed in male mice is likely due to increased viral loads and infiltrated immune cells in the lungs. Interestingly, in female mice, CoV-2 levels were significantly reduced in the lungs but significantly increased in the WAT compared to infected male mice. It has been shown that estrogen reduces the levels of ACE2 [50]. Thus, although ACE2 is more highly expressed in adipose tissue than in the lungs [51], females may have reduced levels of ACE2 in WAT because of their higher levels of estrogen. However, SARS-CoV-2 can also infect and invade cells via other receptors and cellular mechanisms [34,52]. For example, SARS-CoV-2 can infect cells through the cholesterol-rich lipid rafts [53,54,55], which may be the case in adipose tissue in female mice. Our data suggested that in female mice adipose tissue may act as a sink/reservoir for SARS-CoV-2 and thus spares the lungs from a greater viral load, preventing pulmonary damage due to infiltrated immune cells and activated pro-inflammatory cytokines. The reduced viral load in the lungs of female mice may also be attributed to an increased pro-inflammatory environment in female mice caused by increased IL-6 and TNF-a levels in adipose tissue, which increases the levels of circulating pro-inflammatory cytokines.

We observed an increased average size of adipocytes in uninfected females compared to uninfected males in the adipose tissue of hACE2 mice, which may be attributed to the lower levels of lipases in female mice. However, the increased levels of lipases in the adipose tissue of infected female mice compared to infected male mice may cause a loss of lipid droplets. CoV-2 infection causes a loss of lipid droplets and promotes cell death in adipose tissue. Our histological and Western blotting analysis demonstrated that the loss of lipid droplets and increased cell death due to lipolysis, necrosis, and apoptosis were significantly higher in the WAT of infected female mice compared to infected male mice. Like other viruses and parasites, SARS-CoV-2 utilizes host lipids for its biosynthetic needs [56]. It has been shown that lipid droplets increase the replication of SARS-CoV-2 [56]. Isolated monocytes from COVID-19 patients showed an increased accumulation of intracellular lipid droplets compared to SARS-CoV-2 negative donors [56], suggesting that CoV-2 manipulates cellular metabolism to acquire lipid resources from the host. Thus, adipocytes, which are rich in lipid droplets, provide the necessary fuel for viral replication. In female hACE2 mice, the presence of CoV-2 in adipose tissue increased the loss of lipid droplets and caused cell death, which likely resulted in the infiltration of immune cells and the elevation of cytokines such as IL-6 and TNF-α. The difference in viral load and immune cell activation can be attributed to lipid droplets. The loss of lipid droplets due to deregulated lipolysis has been linked to the infiltration of immune cells, immune cell activation, and cell death (apoptotic or necrotic) [57,58,59,60]. The process of cell death initiates the infiltration of immune cells and the release of TNFα, which in turn further elevates lipolysis [61]. In general, male mice have more fat compared to female mice and male mice are more susceptible to developing obesity [62,63]. However, the levels of body fat in hACE2 mice were not measured. These basic metabolic differences in fat tissue between males and females likely contribute to the greater levels of CoV-2 in females, leading to higher levels of pro-inflammatory cytokines in adipose tissue. The cytokines released in adipose tissue contribute to the elevated cytokine levels in circulation [64]; thus, in infected female mice, TNF-α and IL-6 released from the adipose tissue could activate immune cells and contribute to the observed reduction of viral load in the lungs [65,66].

## 4. Materials and Methods

### 4.1. Biosafety

All aspects of this study were approved by the Institutional Animal Care and Use Committee (IACUC) and Institutional Biosafety Committee of Center for Discovery and Innovation (CDI)—Hackensack University Medical Center and adhered to the National Research Council guidelines.

### 4.2. Animal Model and Experimental Design

The transgenic mice expressing human angiotensin-converting enzyme 2 (hACE2) were purchased from Jackson Laboratories, Bar Harbor, ME and bred at CDI animal research facility. Both male and female mice (*n* = 8) were intra-nasally infected with 1 × 10^4^ pfu SARS-CoV-2 (NR-52281, Isolate USA-WA1/2020 CoV-2 virus, NIH-BEI Resources, Manassas, VA, USA). After 10 days post infection (DPI), the animals (*n* = 4/sex) were euthanized, and samples such as blood, lungs, spleen, and mesenteric white adipose tissue (WAT) were collected. Age and sex-matched uninfected hACE2 mice served as controls (Appendix A). The lungs and WAT samples alone were used in the present study.

### 4.3. Histological Analysis of Adipose Tissue

Freshly isolated WAT were fixed with 10% neutral-buffered formalin for a minimum of 48 h and then embedded in paraffin wax (*n* = 4/sex). Hematoxylin and eosin (H&E) and Masson’s trichrome staining were performed, and the images were captured as previously published [46]. Four to six sections of each WAT were analyzed in this study. For each WAT section, the histological evidence of adipose tissue pathology was classified in terms of the presence of infiltrating immune cells, loss of lipid droplets, and fibrosis [46,67].

### 4.4. Determination of SARS-CoV-2 Load in the Tissue

Total RNA was isolated from the lungs and WAT of SARS-CoV-2 infected hACE2 mice using TRIzol reagent. The number of SARS-CoV-2 copies was quantified using a 2019-nCoV_N2 primer/probe mix and One-Step PrimeScript RT-PCR kit (Takara Bio Inc., San Jose, CA, USA). All assays were performed on Agilent AriaMx Real-time PCR System according to the following cycling conditions: 15 min at 42 °C (1 cycle, reverse transcription), followed by 10 sec at 95 °C (1 cycle, hot start) and continuing with 5 sec at 95 °C, and 30 sec at 55 °C (40 cycles, PCR amplification).

### 4.5. Immunohistochemistry Analysis of SARS-CoV-2

Freshly isolated WAT tissues were fixed with 10% neutral-buffered formalin for a minimum of 48 h and then embedded in paraffin wax (*n* = 4/sex) and sectioned for immunohistochemistry analysis (IHC). IHC was performed using a rabbit monoclonal anti-SARS-CoV-2 nucleocapsid protein (#NR-53791, Sino Biological, Wayne, PA, USA) with a dilution of 1:1000 followed by biotinylated secondary antibody using VECTASTAIN Elite ABC-HRP kit (#PK-6101, Vector Laboratories, Newark, CA, USA). The sections were then washed and incubated with peroxidase substrate and counterstained with hematoxylin. 

### 4.6. Immunoblot Analysis

Tissue lysates were prepared as previously described [30]. The protein concentration quantitation was performed using a Pierce BCA protein assay kit (#23225, ThermoFisher Scientific, Waltham, MA, USA). Then, 30 µg total protein from each sample was loaded and resolved on 8% or 15% SDS-PAGE as appropriate and transferred onto nitrocellulose membrane for immunoblot analysis. Primary antibodies against CD4 (#NBP1-19371, Novus Biologicals, Centennial, CO, USA), CD8 (#NBP2-29475, Novus Biologicals, Centennial, CO, USA), F4/80 (#NB 600-404, Novus Biologicals, Centennial, CO, USA), TNFα (#ab6671, Abcam, Waltham, MA, USA), IL-6 (#66146-1-lg, Proteintech, Rosemont, IL, USA), IL-10 (#20850-1-AP, Proteintech, Rosemont, IL, USA), ATGL (#2439, Cell Signaling Technology, Danvers, MA, USA), HSL (#4139, Cell Signaling Technology, Danvers, MA, USA), caspase 3 (#9662, Cell Signaling Technology, Danvers, MA, USA), and BNIP3 (#44060, Cell Signaling Technology, Danvers, MA, USA) were used to detect the expression of corresponding proteins. Horseradish peroxidase (HRP)-conjugated anti-mouse immunoglobulin (#7076, Cell Signaling Technology, Danvers, MA, USA), HRP-conjugated anti-rabbit immunoglobulin (#7074, Cell Signaling Technology, Danvers, MA, USA), and HRP-conjugated anti-rat immunoglobulin (#112-035-003, Jackson ImmunoResearch, West Grove, PA, USA) were used as appropriate secondary antibodies to detect chemiluminescent signal using Invitrogen iBright Imaging Systems. Β-Actin (#4967, Cell Signaling Technology, Danvers, MA, USA) and Guanosine nucleotide dissociation inhibitor (GDI) (#71-0300, Invitrogen, Waltham, MA, USA) were used as appropriate control to assess equal protein loading.

### 4.7. Statistical Analysis

Data represent means ± S.E. Data were pooled, and statistical analysis was performed on GraphPad Prism software version 9.4.1 using two-way ANOVA and Student’s *t*-test as appropriate and significant differences were determined as *p* values between <0.0001 and <0.05 as appropriate.

## 5. Conclusions

In conclusion, our studies suggest that CoV-2 infection affects adipose tissue physiology which could alter systemic metabolic and immune homeostasis during COVID-19. It will be of great importance to further investigate the link between adipose tissue pathophysiology and pulmonary viral load and COVID-19 severity in COVID-19 research. Thus, further mechanistic studies are warranted to understand the role of the pathophysiology of adipose tissue in the pathogenesis of CoV-2 infection and COVID-19 outcomes. Further studies may determine the mechanistic roles of various fat tissues in regulating immune and metabolic signaling in male and female COVID-19 patients.

## Figures and Tables

**Figure 1 ijms-24-01314-f001:**
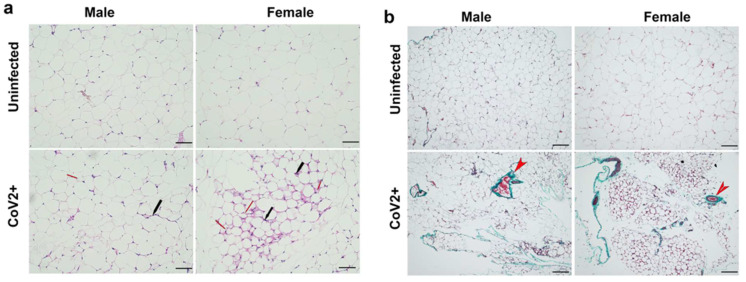
SARS-CoV-2 infection alters adipose tissue morphology at 10 DPI in hACE2 mice (*n* = 4 mice/sex). (**a**) H&E-stained sections of WAT of both males and females showing infiltrated immune cells (black arrow) and loss in lipid droplets (red arrow, smaller adipocytes) with CoV-2 infection compared to uninfected mice (20× magnification, scale bar—50 µm); (**b**) Masson-trichrome staining of WAT sections showing fibrosis and collagen (green color, red arrowhead) in CoV-2 infected mice (10× magnification, scale bar—100 µm).

**Figure 2 ijms-24-01314-f002:**
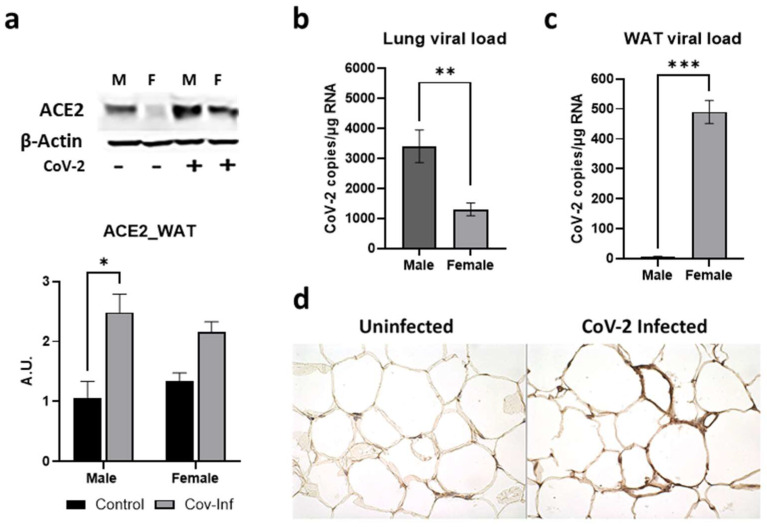
Alterations in ACE2 levels and SARS-CoV-2 viral load in the WAT of male and female CoV-2 infected mice. (**a**) Immunoblot analysis of ACE2 in WAT (top). β-Actin was used as loading control. Fold changes in the protein levels of ACE2 were normalized to β-Actin expression and is shown as a bar graph (bottom). The error bars represent standard error of the mean. * *p* < 0.05 compared to uninfected sex-matched mice (*n* = 4/sex/group) (M, male; F, female); Real-Time PCR analysis showing the number of CoV-2 viral copies/µg of RNA in the (**b**) lungs and (**c**) WATs of male and female CoV-2 infected mice. The error bars represent standard error of the mean (** *p* ≤ 0.01 and *** *p* ≤ 0.001); (**d**) IHC staining of anti-SARS-CoV-2 nucleocapsid protein showing the presence of CoV-2 in adipose tissue around the lipid droplets of infected female hACE2 mice (*n* = 4 mice, a minimum number of five images/section were analyzed).

**Figure 3 ijms-24-01314-f003:**
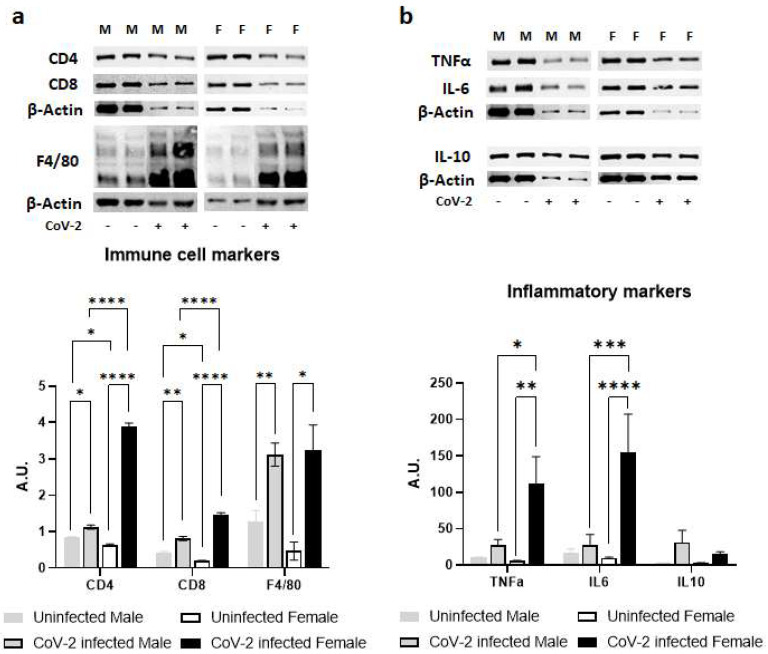
Immune signaling in the WAT during CoV-2 infection is altered differently between the sexes. Immunoblot images of (**a**) immune cell markers (CD4, CD8, F4/80) and (**b**) inflammatory markers (TNFα, IL-6, IL-10). β-actin was used as loading control. Fold changes in the protein levels were normalized to β-actin expression and are shown as bar graphs. The error bars represent standard error of the mean. * *p* < 0.05, ** *p* < 0.01, *** *p* < 0.001, and **** *p* < 0.0001 compared to uninfected sex-matched mice and between the infected groups (*n* = 4/sex/group) (M, male; F, female).

**Figure 4 ijms-24-01314-f004:**
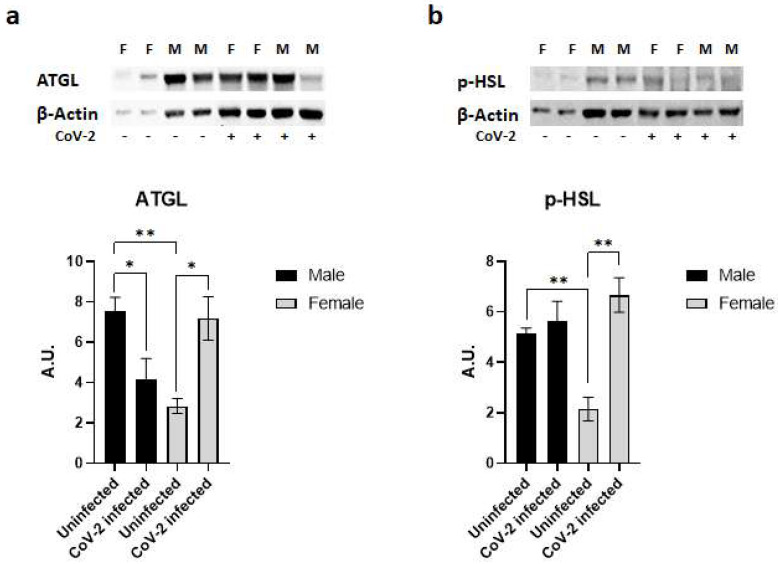
SARS-CoV-2 infection activates lipases in the adipose tissue of female hACE2 mice. Western blot images of (**a**) ATGL and (**b**) p-HSL. β-actin was used as loading control. Fold changes in the protein levels were normalized to β-actin expression and are shown as bar graphs. The error bars represent standard error of the mean. * *p* < 0.05 and ** *p* < 0.01 compared to uninfected mice (*n* = 4/sex/group) (M, male; F, female).

**Figure 5 ijms-24-01314-f005:**
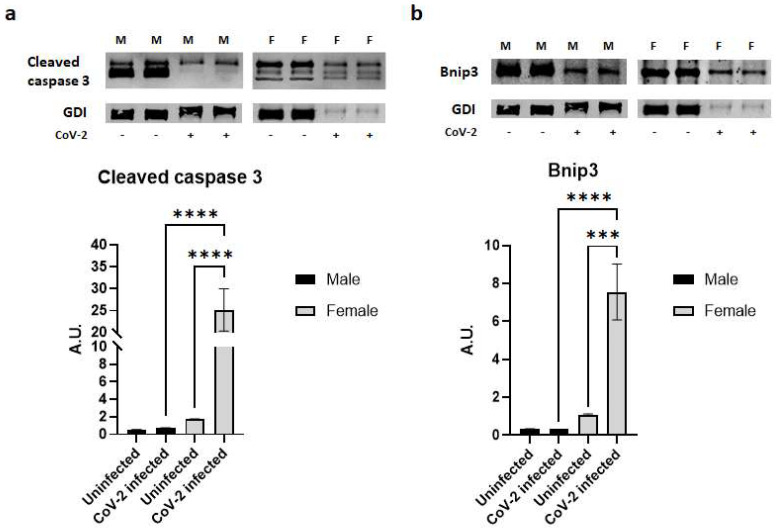
SARS-CoV-2 infection induces severe cell death in adipose tissue in female hACE2 mice. Western blot images of (**a**) Cleaved caspase 3 (apoptosis marker) and (**b**) Bnip3 (necrosis marker). GDI was used as loading control. Fold changes in the protein levels were normalized to GDI expression and are shown as bar graphs. The error bars represent standard error of the mean. *** *p* < 0.001 and **** *p* < 0.0001 compared to uninfected sex-matched mice (*n* = 4/sex/group) (M, male; F, female).

## Data Availability

Not applicable.

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
