# Peer review of "Susceptibility of Fat Tissue to SARS-CoV-2 Infection in Female hACE2 Mouse Model"

_ijms, 2023, doi:10.3390/ijms24021314_

Round 1
Reviewer 1 Report
In this manuscript, Thangavel et al use an humanized ACE2 mouse model to study the effect of SARS-CoV-2 infection over the adipose tissue. They compared males and females and conclude that there is sexual dimorphism in regard to adipose tissue viral infection. The research question brought forth by the authors is a relevant one, but the model is underexplored, some of the conclusions are not sufficiently supported by data and there are several major concerns the authors need to address before publication. Please see these concerns below.
Major concerns
1) The authors choose to study mesenteric fat. Why? There is solid literature showing different degrees of susceptibility by different adipose tissue depots to SARS-CoV-2 infection. The authors lose the opportunity to further characterize these differences in vivo.
2) The phenomenon of body fat loss should be better characterized. The authors should provide data on fat mass, lean mass and food intake. I am not entirely convinced that adipose tissue infection is the only cause for what has been mentioned as a loss of body fat. In fact, the authors could do a much more thorough characterization of adipose tissue function and whole body metabolism in these mice.
3) The authors attribute a decrease in fat mass to a “loss of lipid droplets” related to increased adipocyte cell death. That is a possibility, but other possibilities should be entertained, such as altered lipolysis, lipogenesis, and/or adipogenesis. The authors should measure markers of those processes.
4) The immunohistochemistry analysis focus on adipocytes. Are there other cell types infected? To what level adipocyte infection accounts for adipose tissue viral load?
5) Figure 1: images should be improved. Also, the tissues look much more fibrotic in b than in a; perhaps the images are even at a lower magnitude, despite the same scale bar. What is the reason for such differences? Moreover, in b most fibrotic regions are around blood vessels in the CoV2+ samples, including the regions pointed by the arrows. The authors should show regions with blood vessels in the uninfected group too.
6) The authors claim that “adipose tissue in females may act as a sink/reservoir for CoV-2” (page 4, line 112) and “female mice adipose tissue acts as a sink/reservoir for SARS-CoV-2 and thus spares the lungs from a greater viral load, preventing pulmonary damage due to infiltrated immune cells.” (page 9, line 209) – there is not sufficient evidence for that claim.
7) Why do the authors think female fat is more susceptible to SARS-Cov-2 if it expresses less ACE2? They should at least discuss it.
8) Figures 3 and 4: The quantification plots do not reflect the blots as most bands appear to be decreased in infected tissues, including GDI used as the loading control. Also, the differences between males and females are not clear. This should be carefully revised as it sustains important claims of the manuscript. Importantly, a proper loading control should be used.
9) Conclusions: “In conclusion our studies suggest that adipose tissue play a major role in regulating viral load, pulmonary pathology and immune response during SARS-CoV-2 infection which may differ between the sexes.” – This is an overstatement. There is not sufficient data for the authors to conclude that adipose tissue plays a major role in regulating viral load, pulmonary pathology and immune response during SARS-CoV-2 infection.
Minor concerns
1) In page 4, line 113: “inverse relationship exists between the viral loads in the lungs and adipose tissue.” and page 7, lines 199-200: “an inverse relationship exists between the viral loads in the lungs and adipose tissue, and it differs between the males and females” – these sentences need to be rephrased to make it clear that the inverse relationship occurs when comparing males and females. We don’t know if such inverse relationship is universal.
2) Figure 2d: Are these images from males or females?
3) In some occasions the authors claim novelty on aspects that have been demonstrated by others. For example, when they state in page 7, lines 197-199: “Our study revealed for the first time that: (a) SARS-CoV-2 infects adipose tissue and the adipose tissue viral load differs between males and females;” - the first papers showing that SARS-CoV-2 infects adipose tissue should be cited.
4) Page 8, line 224-225: “The difference in viral load and immune cell activation can be attributed to lipid droplets.” – In what basis the authors conclude that?
5) Page 8, line 233-234: “…in infected female mice TNF-α and IL-6 released from the adipose tissue may contribute to the observed reduction of viral load in the lungs.” – this is too speculative.
Author Response
We thank the reviewers for their evaluation and comments. We would like to inform the editor as well as the reviewers that the manuscript submitted is a short communication and not a full-length article. The reviewers suggested for major and minor revisions. We have addressed the concerns of the reviewers in our revised manuscript by providing additional data, revising figures, and adding supplemental information. With the additional data in the revised manuscript, the editor can decide whether this can be a full-length article or a short communication. The reviewers’ comments have further strengthened our work and clarified our results. A detailed response to the reviewers' suggested modifications and corrections follows. The reviewers’ comments are in italicized size 10 font.
Reviewer# 1’s Comments and Authors’ Response
In this manuscript, Thangavel et al use an humanized ACE2 mouse model to study the effect of SARS-CoV-2 infection over the adipose tissue. They compared males and females and conclude that there is sexual dimorphism in regard to adipose tissue viral infection. The research question brought forth by the authors is a relevant one, but the model is underexplored, some of the conclusions are not sufficiently supported by data and there are several major concerns the authors need to address before publication. Please see these concerns below.
Major concerns
- The authors choose to study mesenteric fat. Why? There is solid literature showing different degrees of susceptibility by different adipose tissue depots to SARS-CoV-2 infection. The authors lose the opportunity to further characterize these differences in vivo.
Response: Mesenteric fat is the most analogous to human intra-abdominal tissue both in its location and biology (PMC 4835715). During infections, mice tend to lose body fat. In general, the amount of RNA and proteins are very low in fat tissues. Because the amount of mesenteric fat is greater compared to other depots in infected mice, the fat depots surrounding the intestine (mesenteric) provide sufficient material to perform histology, qPCR analysis, and protein analysis. We agree with the reviewer that we lost the opportunity to further characterize the effect of SARS-CoV2 infection on various fat depots since we had not collected different adipose tissue depots.
- The phenomenon of body fat loss should be better characterized. The authors should provide data on fat mass, lean mass and food intake. I am not entirely convinced that adipose tissue infection is the only cause for what has been mentioned as a loss of body fat. In fact, the authors could do a much more thorough characterization of adipose tissue function and whole body metabolism in these mice.
Response: We did not measure body fat mass, lean mass and food intake in these mice since the COVID-infected mice experiments were performed in a BSL3 facility. However, we have measured the protein levels of lipases in the tissue lysates to demonstrate the effect of infection on lipolysis and fat loss during infection. The new data (Fig. 4) are included in the revised manuscript.
- The authors attribute a decrease in fat mass to a “loss of lipid droplets” related to increased adipocyte cell death. That is a possibility, but other possibilities should be entertained, such as altered lipolysis, lipogenesis, and/or adipogenesis. The authors should measure markers of those processes.
Response: As suggested by the reviewer, we measured the protein levels of lipases in the tissue lysates to demonstrate the effect of infection on lipolysis and fat loss during infection. The new data (Fig. 4) are included in the revised manuscript.
- The immunohistochemistry analysis focus on adipocytes. Are there other cell types infected? To what level adipocyte infection accounts for adipose tissue viral load?
Response: We did not analyze whether other cells (besides adipocytes) in adipose tissue were infected by SARS-CoV2. Immunohistochemistry analysis showed the presence of viral protein adjacent to lipid droplets in the sections of adipose tissue.
- Figure 1: images should be improved. Also, the tissues look much more fibrotic in b than in a; perhaps the images are even at a lower magnitude, despite the same scale bar. What is the reason for such differences? Moreover, in b most fibrotic regions are around blood vessels in the CoV2+ samples, including the regions pointed by the arrows. The authors should show regions with blood vessels in the uninfected group too.
Response: The resolution of the images presented in Fig. 1 was 500 dpi. Because they were inserted in a word file, the images might have lost resolution. We also submitted the figures as TIFF files (supporting data). The images in Figure 1a and Figure1b look different and Fig 1b looks more fibrotic because the staining used in Fig 1b was to detect fibrosis (Masson-trichrome staining). However, in Figure 1a, the tissue sections were stained with Hematoxylin and eosin. We observed very low or no fibrosis in uninfected fat tissue compared to infected mice. We apologize for the typo in Figure 1b legend. The scale bar should be 100um. We corrected the error in the revised manuscript.
- The authors claim that “adipose tissue in females may act as a sink/reservoir for CoV-2” (page 4, line 112) and “female mice adipose tissue acts as a sink/reservoir for SARS-CoV-2 and thus spares the lungs from a greater viral load, preventing pulmonary damage due to infiltrated immune cells.” (page 9, line 209) – there is not sufficient evidence for that claim.
Response: Based on our previous work in different infectious disease models, we think that adipose tissue may also serve as a reservoir for CoV2. Adipose tissue in female mice may provide a better environment for the survival of pathogens because adipose tissue biology and metabolic signaling differ between the sexes [PMC6525964; PMC3756100], influencing their responses to infection. We added an explanation to support our conclusion in the revised manuscript. The reduced viral load in the lungs in female mice may also be attributed to a more pro-inflammatory environment in female mice caused by increased IL-6 and TNF-a levels in adipose tissue, which increases the levels of circulating pro-inflammatory cytokines.
- Why do the authors think female fat is more susceptible to SARS-Cov-2 if it expresses less ACE2? They should at least discuss it.
Response: It has been shown that female hormone estrogen reduces the levels of ACE2. Adipose tissue in females is the main source of ACE2. This could be the cause for reduced ACE2 levels in female mice. However, SARS-CoV2 can also infect and invade cells via other receptors and cellular mechanism(s). For examples, SARS-CoV2 can infect cells through the cholesterol rich lipid rafts. Thus, it is likely that in adipose tissue in female mice SARS-CoV2 invades adipocytes via lipid drafts. Our histology sections demonstrated that adipocytes were on average larger in female mice compared to male mice. We have discussed the role of lipid rafts in adipocytes as a possible invasion route for CoV2 infection in the revised manuscript.
- Figures 3 and 4: The quantification plots do not reflect the blots as most bands appear to be decreased in infected tissues, including GDI used as the loading control. Also, the differences between males and females are not clear. This should be carefully revised as it sustains important claims of the manuscript. Importantly, a proper loading control should be used.
Response: GDI (guanine nucleotide dissociation inhibitor) protein has been used as a loading control in many studies. However, as suggested by the reviewer, we also used b-actin and GAPDH as loading controls. Figures in the revised manuscript include the images of b-actin or GAPDH as loading controls.
- Conclusions: “In conclusion our studies suggest that adipose tissue play a major role in regulating viral load, pulmonary pathology and immune response during SARS-CoV-2 infection which may differ between the sexes.” – This is an overstatement. There is not sufficient data for the authors to conclude that adipose tissue plays a major role in regulating viral load, pulmonary pathology and immune response during SARS-CoV-2 infection.
Response: We rephrased the conclusion part as follows: In conclusion, our studies suggest that adipose tissue may play a role in regulating pulmonary viral burdens by altering systemic metabolic and immune homeostasis during COVID. Thus, further mechanistic studies are warranted to understand the role of the pathophysiology of adipose tissue in the pathogenesis of CoV2 infection and COVID outcomes. Further studies may determine the mechanistic roles of various fat tissues in regulating immune and metabolic signaling in male and female COVID patients.
Minor concerns
- In page 4, line 113: “inverse relationship exists between the viral loads in the lungs and adipose tissue.” and page 7, lines 199-200: “an inverse relationship exists between the viral loads in the lungs and adipose tissue, and it differs between the males and females” – these sentences need to be rephrased to make it clear that the inverse relationship occurs when comparing males and females. We don’t know if such inverse relationship is universal.
Response: Our data demonstrated an inverse relationship between the viral loads in the lungs and adipose tissue, and it differed between the male and female huACE2 mice. However, we do not know if this correlation is universal. We revised those sentences in the revised manuscript.
- Figure 2d: Are these images from males or females?
Response: These images were from female mice. We included the information in Fig. 2d legend in the revised manuscript.
- In some occasions the authors claim novelty on aspects that have been demonstrated by others. For example, when they state in page 7, lines 197-199: “Our study revealed for the first time that: (a) SARS-CoV-2 infects adipose tissue and the adipose tissue viral load differs between males and females;” - the first papers showing that SARS-CoV-2 infects adipose tissue should be cited.
Response: As suggested by the reviewer, we revised the manuscript as follows: “the viral loads in adipose tissue and the lungs differ between males and females in hACE2 mice”.
- Page 8, line 224-225: “The difference in viral load and immune cell activation can be attributed to lipid droplets.” – In what basis the authors conclude that?
Response: We included citations to support that the loss of lipid droplets/increased lipolysis causes infiltration and activation of immune cells. In addition, lipolysis may also provide the required fatty acids for the survival and/or replication of pathogens in adipose tissue. We added relevant citations to support these statements in the revised manuscript.
- Page 8, line 233-234: “…in infected female mice TNF-α and IL-6 released from the adipose tissue may contribute to the observed reduction of viral load in the lungs.” – this is too speculative.
Response: It has been shown that adipose tissue-derived inflammatory mediators, including cytokines, can regulate systemic immune homeostasis by altering circulating cytokine levels. We added relevant citations to support this statement in the revised manuscript.
Reviewer 2 Report
The author conducted a very important study to identify the different susceptibility of respiratory infectious diseases by gender. I enjoyed reading this manuscript. But the following questions should be clearly explained.
The author explained that ACE2 transports the COVID virus into the cells. ACE2 in WAT significantly increased in males, more than in females, but the WAT viral load in males is almost flat and much less than in females.
The author explained that the viral loads in lung and adipose tissue showed an inverse relationship. Usually, the body fat tissue content is higher in females, then adipose tissue has a higher viral load and lung viral load is less, indicating less severity in females. In this case, the authors' argument makes sense. However, the author cited that male mice had more fat in the discussion.
During the COVID, we learned that the COVID severity was higher in obese people. How can this be explained, considering the authors' findings?
If these can be clearly explained, the quality of the presentation would be improved.
Author Response
We thank the reviewers for their evaluation and comments. We would like to inform the editor as well as the reviewers that the manuscript submitted is a short communication and not a full-length article. The reviewers suggested for major and minor revisions. We have addressed the concerns of the reviewers in our revised manuscript by providing additional data, revising figures, and adding supplemental information. With the additional data in the revised manuscript, the editor can decide whether this can be a full-length article or a short communication. The reviewers’ comments have further strengthened our work and clarified our results. A detailed response to the reviewers' suggested modifications and corrections follows. The reviewers’ comments are in italicized size 10 font.
Reviewer# 2’s Comments and Authors’ Response
- The author conducted a very important study to identify the different susceptibility of respiratory infectious diseases by gender. I enjoyed reading this manuscript. But the following questions should be clearly explained.
The author explained that ACE2 transports the COVID virus into the cells. ACE2 in WAT significantly increased in males, more than in females, but the WAT viral load in males is almost flat and much less than in females. Response: It has been shown that estrogen reduces the levels of ACE2 and that in females is the main source of ACE2 is adipose tissue. This could be the cause of reduced ACE2 levels in female mice. However, SARS-CoV2 can also infect and invade cells via other receptors and cellular mechanism(s). For example, SARS-Cov2 can infect cells through the cholesterol-rich lipid rafts. Thus, it is likely that in the adipose tissue of female mice SARS-CoV2 invades adipocytes via lipid drafts. Our histology sections demonstrated that the adipocytes were on average larger in female mice than in male mice. We discuss the role of lipid rafts in adipocytes as a possible invasion route for CoV2 infection in the revised manuscript.
- The author explained that the viral loads in lung and adipose tissue showed an inverse relationship. Usually, the body fat tissue content is higher in females, then adipose tissue has a higher viral load and lung viral load is less, indicating less severity in females. In this case, the authors' argument makes sense. However, the author cited that male mouse had more fat in the discussion.
Response: In general, male mice had more fat compared to female mice, unlike in humans. However, we do not know the levels of body fat in hACE2transgenic mice. Interestingly, H&E sections of adipose tissue demonstrated that the size of adipocytes (lipid droplets) was slightly bigger in female mice compared to male mice (Fig. 1a). The size of adipocytes, the composition of lipid droplets and lipid rafts, and the sex hormones may all regulate the load in adipose tissue. We discuss these possibilities in the revised manuscript.
- During the COVID, we learned that the COVID severity was higher in obese people. How can this be explained, considering the authors' findings?
Response: Although during the early period of COVID it was thought that obese individuals are at high risk for COVID severity, it was the dysfunctional immune system (due to diabetes and metabolic disorders) associated with obesity that increased the risk for COVID severity, whereas people with higher BMI (increased adiposity) without metabolic disorders likely be more protected. [https://www.thelancet.com/journals/landia/article/PIIS2213-8587(22)00158-9/fulltext]. Further studies need to be conducted to understand the mechanistic link between BMI, obesity, diabetes, and the risk to COVID.
- If these can be clearly explained, the quality of the presentation would be improved.
Response: As suggested by Reviewer#2, we revised the manuscript by adding the above explanations in the discussion part to improve the quality of the manuscript.
Round 2
Reviewer 1 Report
The authors have partially addressed my concerns, but I still find some of the conclusions of the manuscript very speculative and not supported by data. For example, I am still not convinced that adipose tissue infection is the main cause for what has been described (though not shown) as a loss of body fat in hACE2 mice upon SARS-CoV-2 infection. I am also not convinced of the cause of the structural changes occurring in the adipose tissue of infected animals, if they are directly linked to adipose tissue infection or a consequence of metabolic changes at the systemic level, if they are caused by changes in adipogenesis, lipolysis, inflammation and if these features are causally linked, etc. The link between lung infection and adipose tissue infection is also loose. Overall, the conclusions continue to be overstated. I understand this is a short report and I would be happy if the authors toned down their claims and be more contained with their conclusions, but there is a major limitation of the manuscript that needs to be addressed in any case: I am not at all convinced of the western blotting data presented in the manuscript. The loading controls vary greatly and the quantification do not support what the images show. Finally, I asked the authors to show regions with blood vessels in the uninfected group in Fig. 1b because it is clear that the most fibrotic regions are around the blood vessels. If no blood vessels are shown in the uninfected controls, this is an unfair comparison. I recommend the authors to show more images in any case.
Author Response
We thank the reviewers for their evaluation and comments. We would like to inform the editor as well as the reviewers that the manuscript submitted is a short communication and not a full-length article. The reviewers suggested for major and minor revisions. We have addressed the concerns of the reviewers in our revised manuscript by providing additional data, revising figures, and adding supplemental information. With the additional data in the revised manuscript, the editor can decide whether this can be a full-length article or a short communication. The reviewers’ comments have further strengthened our work and clarified our results. A detailed response to the reviewers' suggested modifications and corrections follows. The reviewers’ comments are in italicized font.
Reviewer #1: Comments and Suggestions for Authors
The authors have partially addressed my concerns, but I still find some of the conclusions of the manuscript very speculative and not supported by data. For example, I am still not convinced that adipose tissue infection is the main cause for what has been described (though not shown) as a loss of body fat in hACE2 mice upon SARS-CoV-2 infection. I am also not convinced of the cause of the structural changes occurring in the adipose tissue of infected animals, if they are directly linked to adipose tissue infection or a consequence of metabolic changes at the systemic level, if they are caused by changes in adipogenesis, lipolysis, inflammation and if these features are causally linked, etc. The link between lung infection and adipose tissue infection is also loose. Overall, the conclusions continue to be overstated. I understand this is a short report and I would be happy if the authors toned down their claims and be more contained with their conclusions, but there is a major limitation of the manuscript that needs to be addressed in any case: I am not at all convinced of the western blotting data presented in the manuscript. The loading controls vary greatly, and the quantification do not support what the images show. Finally, I asked the authors to show regions with blood vessels in the uninfected group in Fig. 1b because it is clear that the most fibrotic regions are around the blood vessels. If no blood vessels are shown in the uninfected controls, this is an unfair comparison. I recommend the authors to show more images in any case.
Response:
We understand the concerns of Reviewer#1. In the revised manuscript, we rephrased the conclusion part and toned down the claims as suggested by the reviewer (Please see the conclusion section). My laboratory has been investigating the interlinked role of adipose tissue to various pathogens and the pathogenesis of infectious diseases for more than 15 years. Previously we demonstrated that Trypanosoma cruzi, a parasite that causes Chagas heart disease infects adipose tissue and persists in various depots of adipose tissue [PMIDs: 15843370, 21726660, 22293433, 33826636]. Using magnetic resonance imaging we performed a detailed body composition analysis and showed that loss of body fat was correlated with cardiomyopathy [PMID: 15843370]. We also showed that a high-fat diet increased body fat and protected the heart from acute T. cruzi infection [PMIDs: 25275627, 30071300]. We previously demonstrated that Mycobacterium tuberculosis which causes tuberculosis can persist in adipose tissue in an aerosol-infected murine TB model [PMIDs: 29109018, 30992360]. We showed that loss of body fat increases bacterial load and pathology in the lungs of infected mice using fat ablatable FAT-ATTAC mice [PMID: 30992360]. However, in the case of COVID studies, we used hACE2 mice in a BSL3 facility and we didn’t have any access to perform whole-body MRI, PIXIMUS, or able to induce fat loss to directly correlate with the effect of altered adipose tissue physiology on pulmonary pathogenesis or viral load. In fact, we observed mice not eating a normal amount of food at the peak of infection (3–4-day post-infection). However, the mice we analyzed and presented data in this manuscript are at 10-day post-infection and they were mobile, active, and eating normally at 10 DPI.
We appreciate and thank the reviewer for detecting that the band intensities and the representing bar graphs were not matching in the western blotting figures. When we changed the loading controls in the figures from GDI to β-actin during resubmission, we missed adding the appropriate β-actin images (loading control) for CD4 and CD8 images and TNFa and IL-6 images (Fig. 3a). It was a technical mistake, and we apologize to the reviewer for the confusion it caused in the western blotting figures. We rechecked all the figures and corrected with appropriate images for loading controls. We reused each blot to probe 4-5 target proteins including two loading controls. We replaced the loading controls to β-actin in all the figures in the revised manuscript. Please see below the Fig showing the raw images of the blot stained with ponceau, and probed for β-actin, GDI, caspase 3 and Bnip3 (F-female; M-male). The arrows in the figure shows the correct protein band. The protein levels in adipose tissue lysate of infected mice were apparently lower compared to uninfected group as indicated by ponceau and loading controls (β-actin and GDI).
As the reviewer recommended, we have added a new supplemental figure (Supplemental Fig. 2) showing the regions with blood vessels in the uninfected group in support of Fig. 1b.

Reviewer 2 Report
I am satisfied with the authors' responses and revisions. Although more evidence is necessary to be conclusive, this is a new study on the link between COVID-19 and the role of adipose tissue. I hope the authors keep working on this topic to confirm their findings.
Author Response
We thank the reviewers for their evaluation and comments. We would like to inform the editor as well as the reviewers that the manuscript submitted is a short communication and not a full-length article. The reviewers suggested for major and minor revisions. We have addressed the concerns of the reviewers in our revised manuscript by providing additional data, revising figures, and adding supplemental information. With the additional data in the revised manuscript, the editor can decide whether this can be a full-length article or a short communication. The reviewers’ comments have further strengthened our work and clarified our results. A detailed response to the reviewers' suggested modifications and corrections follows. The reviewers’ comments are in italicized font.
Reviewer#2 Comments and Suggestions for Authors
I am satisfied with the authors' responses and revisions. Although more evidence is necessary to be conclusive, this is a new study on the link between COVID-19 and the role of adipose tissue. I hope the authors keep working on this topic to confirm their findings.
We thank the reviewer for the encouraging suggestions. We will continue our investigations to understand the link between adipose tissue pathophysiology and pulmonary COVID severity and outcome.
Round 3
Reviewer 1 Report
The authors have addressed most of my concerns but haven’t satisfactorily addressed an important one: my concern about the loading controls in the western blotting. They corrected some of the images in the revised manuscript but still, there are some remarkable differences in the beta-actin and GDI levels across groups. Just to give an example that stands out among several others, in Fig.5a, the levels of beta-actin in the female samples go from abundant to undetected after SARS-CoV-2 infection. Obviously, because the denominator is very low (I even wonder how the authors managed to perform the normalization when there are no detectable bands), the levels of the cleaved caspase 3 appear much higher in infected females. The differences in loading are also noticeable in the ponceau, but less so. To me, this shows that there is either major problems with the loading of the western blots or the loading controls are not appropriate. This is important because some of the main conclusions of the manuscript depend on the interpretation of the western blots.
Author Response
Response: We agree with the reviewer that it is difficult to perform protein normalization if there are no detectable bands on western blots because the densitometry values would be very low. As we have shown in our raw data (also in the previous response letter to the reviewer) the levels of b-actin in some of the lanes are not detectable (for example Fig. 5a, female samples). Therefore, we used GDI which is highly expressed in adipose tissue as a loading control to normalize the protein levels and presented the figures with GDI bands in our first submission. We agree that b-actin and GAPDH are the most used loading controls in western blotting analysis. However, GDI has been used as a loading control in western blotting analysis in our laboratory and also in other laboratories. We have published more than 20 research articles in reputed journals including PloS, Mbio, and JBC in which we have presented western blotting data using GDI as loading controls. Here we are providing some references to the research articles that were published by others indicating GDI is a constitutive marker [J Biol Chem. 2005 Feb 11;280(6):4617-26. doi: 10.1074/jbc.M411863200. Epub 2004 Nov 9. PMID: 15536073] and a good loading control in the western blotting analysis [PMIDs: 10823826, 17717599, 15841211, 14576179] including a recent research article in Nature communication [PMID: 34083525]. In the revised manuscript, we revised some of the figures and included GDI as a loading control in some of the figures where b-actin bands are not visible. With this explanation and the references provided, we hope the reviewer's concerns are addressed.
Round 4
Reviewer 1 Report
I understand GDI and beta-actin are two potential good ways to control for loading and normalize western blots. However, they are not universal. They could vary in specific contexts and in the conditions shown in the paper they clearly vary, thus not being suitable to be used as loading controls. Moreover, based on the Pounceau staining, there are some differences in the loading of the western blots. This needs to be adjusted, otherwise the authors will never know if their loading controls are suitable for use or not. Alternatively, they could use Ponceau for loading control and normalization, although the loading still needs to be adjusted in some cases. Finally, the authors need to consider that the linear dynamic range for western blotting quantification is somewhat narrow. If one band is too intense and the other too faint, the quantification is not precise. This reviewer is convinced that if these loading issues are not resolved, the interpretation of the blots is compromised and hence the conclusion of the paper is not sustained.
Author Response
The editor's suggestions are included in the revised manuscript.